# Maleinized Linseed Oil as Epoxy Resin Hardener for Composites with High Bio Content Obtained from Linen Byproducts

**DOI:** 10.3390/polym11020301

**Published:** 2019-02-11

**Authors:** Vicent Fombuena, Roberto Petrucci, Franco Dominici, Amparo Jordá-Vilaplana, Néstor Montanes, Luigi Torre

**Affiliations:** 1Instituto de Tecnología de Materiales (ITM), Universitat Politècnica de València (UPV), Plaza Ferrándiz y Carbonell s/n, 03801 Alcoy, Alicante, Spain; amjorvi@upvnet.upv.es (A.J.-V.); nesmonmu@upvnet.upv.es (N.M.); 2Materials Engineering Center, University of Perugia, Località Pentima Bassa, 21, 05100 Terni, Italy; roberto.petrucci@unipg.it (R.P.); francodominici1@gmail.com (F.D.); luigi.torre@unipg.it (L.T.)

**Keywords:** polymer-matrix composites (PMCs), surface properties, resin transfer molding (RTM), mechanical testing

## Abstract

Green composites, with more than 78 wt.% of products obtained from linen *Linum usitatissimum*, were developed in this research work. Epoxidized linseed oil (ELO) was used as bio-based resin, a mix of nadic methyl anhydride (MNA) and maleinized linseed oil (MLO) were used as cross-linkers and finally, flax fabrics were used to obtain composite laminates by resin transfer molding (RTM). The flax fibers were modified using amino-silane, glycidyl-silane and maleic anhydride treatment in order to increase the compatibility between lignocellulosic fibers and the polymeric matrix. Mechanical and thermal properties were studied by flexural, tensile and impact test, as well as dynamic mechanical analyses (DMA) to study the viscoelastic behavior. Contrary to what could be expected, when fibers are previously treated in presence of MLO, a reduction of anchorage points is obtained causing a substantial increase in the ductile properties compared with composites without previous fiber treatment or without MLO.

## 1. Introduction

*Linum usitatissimum*, also known as linseed or common flax is one of the oldest fiber crops used by mankind [1]. In what is Georgia today, the earliest evidence of the use of flax fibers as textile was found, approximately 30,000 years ago [2]. Later, in ancient Egypt the mummies were wrapped with linen fabrics, where flax was considered as a symbol of purity [3]. In only 125 days the plant is able to complete its life cycle including, vegetative, flowering and maturation periods, which make it a plant of the family of linacea with greater production capacity [2].

In 2017, Canada was the largest producer of flaxseed in the world, representing about 40% of world production (exceeding 2.9 million tons). The joint production of China, the United States, and India represents another 40% of world production [4]. Flax is cultivated for multiple applications, such as, fiber, oil (linseed oil), food supplement, and as an ingredient of wood finishing products. In fact, the Latin term “*usitatissimum*” means “most useful” [5].

Today, due to the growing environmental awareness, the scientific community and manufacturers have focused their attention on flax fiber, as it is the most widely used bio-fiber in composite materials [6]. Flax fibers and other bio-fibers are used as ecological alternatives to glass fiber in composites for engineering applications [7]. In 2010, 13% of the total reinforcement materials were bio-fibers, while a forecasting up to 2020, states that this percentage will be higher than 28% [8]. Bio-fibers, such as flax, have a potential as a raw material in composites used in building industries, sport equipment, packaging, paper, furniture, and automotive industries [9]. The chemical composition of flax fiber, as reported by several authors (about 64%–74% cellulose, 11%–17% hemicellulose, 2%–3% lignin, and 1%–2% pectin, 1.5% waxes, and 8%–10% water) allows an optimal equilibrium between mechanical and ductile properties [10,11].

If flax fibers are compared with E-glass fibers (Table 1), one of the most common fibers used in composites, it is possible to realize that flax fibers have the capability to replace glass. If we also take into account their low price, low toxicity, high strength to weigh ratio, and the possibility of recovering energy at the end of the life cycle, it is clear that flax fibers are highly promising fibers [12,13,14,15].

Moreover, the possibilities of *Linum usitatissimum* in bio-composites do not only comprise its use as reinforcement fiber. The oil obtained from the seed, commonly known as linseed oil, is a triglyceride whose composition depends on the type of plant and growing conditions. Traditionally, this oil has been used as a binder in paint formulations and wood finish due to its drying ability [16]. Despite this, the high unsaturation degree (carbon–carbon double bonds) present in the major fatty acids, such as linoleic and linolenic fatty acids with two and three unsaturations respectively, allows the possibility of converting this vegetable oil into thermosetting resins. To achieve this, polymerization can take place thorough reactions involving these double bonds [17,18]. Although this direct polymerization, (homopolymerization) is feasible and technically viable, the most widely used route is the conversion of double bonds into functional groups that can then be easily polymerized due to an increase in reactivity. This is the case of the epoxidation process, which converts double bonds contained in the different fatty acids into oxirane rings by reaction of the oil with peroxoacids. Epoxidized linseed oil (ELO) has been used in the last decade as bio-based thermoset polymer in multiple studies [16,19,20,21,22].

ELO could be cross-linked with different curing agents, such as dicarboxylic acids [24,25] and cyclic anhydrides. It is important to take into account that basic cyclic anhydrides such as maleic anhydride (MA) and phthalic anhydride (PA) are solids at room temperature and, although they have been used in cross-linking epoxidized vegetable oils [21], it is worthy to note a particular family of cyclic anhydrides which are liquids at room temperature, which is a key feature in using them as curing agents [26]. Cyclic anhydrides such as methylhexahydrophthalic anhydride (MHHPA), methyltetrahydrophthalic anhydride (MTHPA), dodecenyl succinic anhydride (DDSA), methyl nadic anhydride (MNA), among others, are widely used as cross-linking agents in epoxy resins for high thermal performance [16,21,27]. It is necessary to take into account that the curing process of at thermosetting bio-resin with anhydrides is a complex reaction involving different chemical interactions, so that initiators and/or catalysts as imidazole and glycerol respectively are needed. The reactivity of a chemical compound involved in a cross-linking process is measured by the equivalent weight, which indicates the mass of a compound containing an equivalent gram of a reacting chemical group. In epoxy-based materials the equivalent epoxide weight (EEW) is used, while for anhydrides the anhydride equivalent weight (AEW) gives the appropriate information about their ability to form cross-links. Among the liquid cyclic anhydrides, it is worthy to note the increasing use of methyl-5-norbornene-2,3-dicarboxylic anhydride or nadic methyl anhydride (MNA). This petrochemical hardener has an AEW of 178 g equiv^−1^ and its use as a hardener in epoxy resins gives cross-linked epoxy resins which are able to withstand high temperatures [26].

As an environmentally friendly alternative to petrochemical hardeners used in bio-resins, vegetable oils can also be chemically modified to introduce the maleic anhydride in its structure by reacting maleic anhydride (MA) with unsaturations contained in vegetable oils through a combination of Diels-Alder condensation and “ene” reactions. Whang et al. studied the use of maleated castor oil to produce biodegradable foam plastics [28], and Mistri et al. applied maleated castor oil to produce green composites with jute fabrics [29]. Maleated groups present in a chemically modified vegetable oil, can readily react with epoxy groups contained in epoxidized oil in the presence of initiators (polyhydric alcohols) and accelerators (imidazoles). Figure 1 shows the two different pathways to modify the basic triglyceride structure of linseed oil used to give both the base epoxy resin (epoxidized linseed oil, ELO) and maleic anhydride-derived hardener (maleinized linseed oil, MLO). Therefore, chemical compounds derived from vegetable oils are a good alternative to reduce environmental impact, thus contributing to a sustainable development by reducing petroleum dependency and giving a response to the growing demand for greener products [30].

Finally, it is must be taken into account that one of the main drawbacks to manufacture bio-composites using natural fibers, like flax, hemp, jute, kenaf and other lignocellulosic fibers, is the high intrinsic hydrophilicity due to presence of hydroxyl groups in the cellulose structure. This results in the possible formation of ineffective interfaces between the hydrophilic fibers and hydrophobic polymeric matrices. This poor (or lack of) interactions result in debonding problems and presence of voids in the obtained composites [31]. An important method to improve the compatibility along the interface between the fiber and the polymeric matrix is the use of chemical treatments on the fiber. Different functional groups can react with hydroxyl groups in cellulose to minimize the intrinsic hydrophilicity. Among others, it is worthy to remark the use of mercerization sodium hydroxide, and the use of peroxides, organics acids and silanes [10,31,32,33,34,35,36,37] that can block hydroxyl groups. Although there is a wide variety of surface treatments, silanes have been the most widely used due to their effectiveness compared to other methods [38].

This work describes the manufacturing and characterization of high environmental friendly composites using different vegetable oil-derived products from *Linum usitatissimum*. The base resin consisted in an epoxy resin derived from linseed oil (LO), namely, epoxidized linseed oil (ELO). The hardener was partially bio-based as it was composed of petroleum-derived methyl nadic anhydride (MNA) and bio-based maleinized linseed oil (MLO) with a weight fraction of 50:50. Finally, flax fabrics were used as reinforcements and were converted into high environmental efficiency composite materials by resin transfer molding (RTM). The overall bio-based content of these composites is close to 78% (as it can be seen in Figure 2 which shows the flax-derived components). Several surface treatments were carried out on fibers to overcome their intrinsic hydrophilicity. A surface treatment was carried with two different silanes, amino (I) and glycidyl (II) and a third treatment consisted on esterification of hydroxyl groups in cellulose by reaction with maleic anhydride (III).

## 2. Materials and Methods

### 2.1. Materials

As bio-based epoxy matrix an epoxidized linseed oil (ELO) supplied by Traquisa S.A (Barberá del Vallés, Barcelona, Spain) was used. The main parameters of this epoxy resin are a molecular weight of 1037 g mol^−1^, containing ±8% of epoxide oxygen and an average distribution of fatty acids as follows: 3%–5% stearic acid, 5%–7% palmitic acid, 14%–20% linoleic acid, 20%–26% oleic acid and 50%–56% linolenic acid. As a hardener a liquid mixture (50 wt%:50 wt%) from methyl nadic anhydride (MNA) and maleinized linseed oil (MLO) was used. MNA has an anhydride equivalent weight (AEW) of 178 g equiv^−1^ and, it was supplied by Sigma Aldrich (Sigma Aldrich, Madrid, Spain). On the other hand, MLO has an acid value comprised between 105–130 mg KOH g^−1^ and a maximum viscosity at 20 °C of 10 dPa·s. MLO was supplied by Vandeputte Group (Mouscron, Belgium). As accelerator 2 wt.% of 1-methyl imidazole supplied by Sigma Aldrich (Sigma Aldrich, Madrid, Spain) was used. In the same way as the catalyst provider of hydroxyl groups, glycerol (0.8 wt.%) from Sigma Aldrich (Madrid, Spain) was used. These percentages were optimized in previous works [27]. Composites were manufactured with three layers with the same fiber orientation with a surface density of 200 g·m^−2^. Flax fiber was supplied by Hilaturas Ferre (Hilaturas Ferre S.A., Banyeres, Spain). Flax fabrics were subjected to a previous surface treatment to reduce hydrophilicity. Two different silanes and maleic anhydride supplied by Sigma Aldrich (Madrid, Spain) were used. The first silane an aminosilane: [3-(2-aminoethylamine) propyl]-trimethoxysilane while the second silane consisted on a glycidyl silane: [2-(7-oxabicyclo [4.1.0] hept-3-yl) ethyl] silane.

### 2.2. Surface Treatment

Treatment of flax fibers with different silanes was carried out to increase interfacial adhesion between the polymeric matrix and the hydrophilic fiber, thus leading to an improvement of mechanical performances of the resulting fiber/polymer composites. Initially, silanes are hydrolyzed resulting in silanols, which are adsorbed onto the fiber surface, acting as a chemical bridge in the interface after drying which allows chemical bonding of silanol groups with hydroxyl groups in cellulose [39]. Silane treatments were carried out as follows: first a solution of 1 wt.% silane in acetone was prepared and stirred until homogenization. The use of different solvents was broadly studied [38,40,41,42,43,44,45,46,47], however acetone is widely used to promote hydrolysis [31]. Pickering et al. reported that acetone improves the surface roughness of fibers, increasing their specific area [48]. Previous to immersion of flax fabrics into the corresponding silane solution (with amino- or glycidylsilane), pH was adjusted to 3.5 with acetic acid and stirred for 10 min. Then, flax fabrics were immersed in this solution during 1 h and after this, soaked fabrics were dried in an air oven at 65 °C for 12 h. After this, silanized flax fabrics were washed several times with distilled water (until pH = 7 was obtained) and dried in an air oven at 40 °C during 24 h in order to remove residual acetone solution by evaporation.

As an alternative to silanization, a surface treatment with maleic anhydride on flax fabrics was also carried out. This is an effective method to modify the chemical composition of flax fibers and improve the adhesion between the hydrophilic flax fiber and the hydrophobic resin. The procedure proposed by Cantero et al. was followed to carry out the esterification process [49]. Briefly, maleic anhydride was applied in an amount of 10% in respect to the total fibers weight and with a fiber:solvent ratio of 1:25. Fibers were immersed for 24 h in boiling acetone. After this process, fibers were washed with cold water until stabilization of pH was achieved, and finally dried in air recirculating oven at 100 °C for 12 h. Table 2 provides a list of the materials developed in this research and their corresponding labeling.

### 2.3. Composite Manufacturing

The manufacturing process used to develop composite laminates was resin transfer molding (RTM). The equipment consisted of a Hypaject MKII (Plastech, Thermoset Tectonics, Gunnislake, UK). In order to reduce the intrinsic viscosity of the ELO and MNA/MLO hardener and enable the injection of the mixture into the mold, an injection temperature of 60 °C with a pressure of 100 kPa was used. The pressure in the mold cavity was 95 kPa and the selected curing cycle consisted of an isothermal process at 100 °C for 3 h, as optimized in previous works [22]. After the curing process, a demolding temperature of 30 °C was used. All composites were manufactured with the same stacking sequence of 3 layers of flax fabrics with an angle orientation of 0°/90°.

### 2.4. Characterization of Composites

The curing process of the ELO resin, with a mixture of MLO and MNA hardener, was carried out taking into account the main findings of previous works [23]. Dynamic Mechanical Analysis (DMA) was used to study mechanical properties of composites in dynamic conditions [22]. The storage modulus (G′) was determined by using de Hooke’s law for a viscoelastic material:(1)$$G\prime =\frac{\sigma }{\varepsilon }\cos\left(\delta \right)$$
where σ is the applied stress, ε is percentage deformation, and δ is the phase angle between stress and strain. Samples of 40 × 10 × 4 mm^3^ were analyzed in an AR G2 oscillatory rheometer from TA Instruments (New Castle, DE, USA) equipped with a special clamp for solid samples. This clamp system allows evaluating dynamic properties in a combination of torsion and shear. The heating rate was set to 2 °C min^−1^ and the temperature range was programmed from 25 °C to 150 °C using a constant frequency of 1 Hz and a maximum shear deformation (%γ) of 0.1%. As Gernaat et al. described, G′ is an excellent parameter to study the correlation of mechanical properties. It is also useful to evaluate the effect of temperature on mechanical properties [50].

The study of the mechanical properties of the different composites was completed by three different techniques. Tensile tests were carried out following the guidelines of ASTM D3039-08 on a universal testing machine Instron Model 3382. The load cell was 100 kN and a crosshead rate of 2 mm·min^−1^ was used. On the other hand, flexural properties were also studied according the ASTM D790-10, using an universal testing model Lloyd 30 K with a crosshead rate of 1.7 mm·min^−1^ and a load cell of 500 N. Impact toughness of the developed composites was studied by the Charpy impact test on a pendulum model Metrotec S.A. with a maximum energy of 6 J. A minimum of five different samples was characterized and average values for each mechanical property were calculated. All mechanical tests were carried out at room temperature.

The morphology of the fractured samples from impact tests was studied by scanning electron microscopy (SEM) in a Phenom (FEI Company, Eindhoven, The Netherlands) microscope. The coating process lasted 120 s using the model of sputter-coater Emitech mod. SC 7620 (Quorum Technologies, East Sussex, UK). SEM technique was mainly used to identify the fiber–matrix interactions achieved after the different surface treatments on flax fabrics.

## 3. Results and Discussion

In the first part of this study, the mechanical properties, such as tensile and flexural test of the different composites have been exhaustively analyzed (Figure 3). The influence of MLO in the cross-linking mixture and the effects of the different surface treatments are described in this section.

MNA composite is taken as the reference material as it is cross-linked without MLO and, on the other hand, flax fabrics are not subjected to any surface treatment. As can be seen in Figure 3A, the flexural modulus of MNA composite is greater than 6600 MPa. If this value is compared with MNA:MLO composite it is possible to see that the presence of MLO results in a less rigid material (4480 MPa). In the same way, flexural strength of both samples (Figure 3C) suffers a decrease higher than 40%. Different authors such as Ray et al. and Lackinger et al. have reported this decrease in parameters related with resistant mechanical properties [51,52]. This behavior could be attributed to the presence of flexible MLO molecules, which reduce the rigidity of the cross-linked material. Therefore, replacing 50 EEW% of petrochemical hardener as MNA by MLO makes the material greener and more sustainable, but a decrease in mechanical resistant properties is observed while an improvement of ductile properties is expected.

It is possible to observe that, when flax fabrics are previously treated with amino- or glycidyl silane or subjected to a previous esterification with maleic anhydride, mechanical resistant parameters decrease considerably. While MNA composite has a flexural modulus greater than 6600 MPa, composites with flax fabrics treated with amino silane, glycidyl silane and maleic anhydride have values 60.7%, 73.7% and 77.4% lower respectively. The same trend can be observed for flexural strength (Figure 3C), tensile modulus (Figure 3B) and tensile strength (Figure 3D).

These unexpected results are in opposition to what coupling agents or previous esterification usually promote. Multiple studies have reported that the use of surface treatments such as silane coupling agents and maleic anhydride esterification increase the mechanical properties as the hydrophobicity of the lignocellulosic reinforcement decreases so that there is increased affinity with to polymeric matrix [21,31,49,53,54,55]. To elucidate this unusual behavior, an MNA:GLYC composite was manufactured. This composite is characterized by the absence of MLO as co-hardener. Thus, it is possible to determinate that the presence of MLO as a hardener has a direct influence of the behavior of surface-treated flax fabrics. The MNA:GLYC composite has higher mechanical properties than all other composites, including the MNA composite. Thus, this behavior allows obtaining a series of conclusions:

(I) Samples cured with a 50 EEW% of MLO as hardener has greater ductile properties, due to the presence of long and flexible molecules.

(II) When a surface treatment is carried out on fibers, the mechanical resistant properties will be higher, as long as MLO is not used as hardener.

(III) If MLO is used as hardener, surface treatments produce an effect which is opposite to what is expected, leading to a decrease in the mechanical resistant properties.

As indicated previously, all properties related with deformation are increased with MLO presence. It is possible to see that maximum deflection (Figure 3E) and elongation at break (Figure 3F) are clearly higher when 50 EEW% of MLO is added and a previous surface treatment of the flax fabrics is carried out. For this reason, it is possible to confirm a synergistic effect in ductile properties due to the combined effect of the addition of MLO as a co-hardener and previous treatments with silanes, or by maleic anhydride esterification.

Regarding the possible differences of the three surface treatments carried out, it should be noted that the composite material manufactured with MLO and a previous treatment of flax fibers with maleic anhydride, seems to give rise to the material with greater ductile properties. For example, the MNA:MLO:MAL composite has an elongation break 2.7 times higher than MNA composite and 1.9 times higher than MNA:MLO composite. This behavior was previously detailed by other authors, such as Sawpan et al. [31]. However, surface treatment should promote a chain mobility restriction caused by intra- and intermolecular interactions between the polar groups of the epoxy resin and the organic components of the coupling agent [56], but in this case, the presence of MLO leads to the contrary effect. With the aim to summarize this trend on of mechanical and ductile properties a plot representation of tensile strength and% elongation is done in Figure 4.

Regarding impact properties, the main results obtained in the Charpy impact test obtained are summarized in Figure 5. In the first approach, by analyzing the influence of MLO as co-hardener it is possible to conclude that the MNA:MLO composite has an impact energy (kJ·m^−2^) 45.7% higher than the same material using only MNA as hardener (MNA composite). Secondly, previous surface treatments of flax fabrics with amino-, glycidyl silane and anhydride maleic esterification provides increased toughness of around 100.3%, 120.2% and 58.9% respectively with regard to the MNA:MLO sample. The presence of long fatty acid chains in MLO used as partial substitute of MNA (co-hardener) increases considerably the impact properties of the developed composites. Finally, the comparison of results between MNA and MNA:GLYC composites allows demonstrating again that the surface treatment has been carried out in a correct way, and that it is effective only when MLO is not present. When MLO is not used as co-hardener, surface treatment leads to an increment of intermolecular bonding between ELO resin and flax fiber, thus leading to a stiffening effect on the composite (impact energy 1.77 kJ·m^−2^ respect 1.65 kJ·m^−2^). This typical trend has been reported by other authors such as Samper et al. [57].

The dynamic mechanical analysis of the composites has been carried out to study the mechanical properties under dynamic loading and increasing temperature. Figure 6 shows a plot evolution of the storage modulus (G’) as a function of the increasing temperature. Firstly, it is possible to see that DMA curves for composites containing MLO as co-hardener are located below the curve of composites without MLO (MNA and MNA:GLYC). The elastic-vitreous behavior of all composites developed could be detected below the glass transition temperature (*T*_g_), with high storage modulus values. Maximum G′ values are obtained in MNA and MNA:GLYC composites, as expected, with values around 3000 MPa at 30 °C. On the other hand, the increase in ductile properties provided by the use of MLO as co-hardener, can be observed in MNA:MLO composite, manufactured without any surface treatment, which has a G′ of 1000 MPa. Moreover, when a surface treatment is applied on flax fabrics, the storage modulus decreases to values below 350 MPa (in the case of a glycidyl silane treatment). This is in total agreement with previous mechanical results, which suggested a clear softening effect when MLO was used as co-hardener combined with a surface treatment on flax fabrics. Once the glass transition is overpassed the elastic-viscous of the composite provides lower G′ values. It is important to remark that composites without MLO co-hardener offer a sharp decrease, with G′ values around 100 MPa at 100 °C. Meanwhile, the decrease in G′ values for composites containing MLO is much lower, changing from values around 1000 MPa down to 150 MPa, which are even higher than those corresponding to composites without MLO. The influence of the presence of MLO in the mechanical behavior of composites it is considered as a stabilizing effect due to in samples with presence of MLO, the decrease in G′ values are lower, giving as result higher values at high temperatures, in a similar way that has been reported with different byproducts from vegetal oils [29,58,59].

Regarding the phase angle (δ), Figure 7 shows its evolution with increasing temperature. Composites cross-linked with MLO co-hardener provide a shift of the phase angle peak to lower temperatures. The temperature peak of the phase angle represents the glass transition (*T*_g_), which involves the change of behavior from elastic to viscous. MNA and MNA:GLYC composites, both cured only with MNA as hardener, are characterized by a *T*_g_ value of 67.4 °C and 71.7 °C respectively. In absence of MLO co-hardener and no previous surface treatment on flax fabrics give high *T*_g_ values, which is in total agreement with the previous mechanical test. In this case, the use of glycidyl silane provides stronger interaction between flax fiber and the ELO-based epoxy resin. When MLO is used as co-hardener, together with MNA (MNA:MLO composite) the corresponding *T*_g_ value falls down to 52.4 °C, which represents a percentage decrease of about 22,2% with regard to MNA composite. As it has been suggested before, this decrease in *T*_g_ values could be attributed to presence of flexible MLO molecules in composites, which reduce rigidity of the material under dynamic loading conditions [52]. All composites manufactured with a flax fabrics subjected to previous surface treatment, have lower *T*_g_ values, thus giving evidence of a synergistic flexibilization effect by combining MLO co-hardener and previous surface treatments on flax fabrics. The lowest *T*_g_ value is obtained for composites with flax fabrics subjected to silanization with glycidyl silane, with a percentage decrease in *T*_g_ of about 42% with regard to MNA composite. It is important to remark that phase angle peaks in composites cured with MLO as co-hardener, show a broadening effect in a similar way that Chem et al. have reported in polyurethane-modified. This phenomenon could be explained by the decrease in the cross-linking density of the epoxy using a MNA/MLO mixture as hardener, thus leading to a broader distribution and relaxation times of molecules in composite [60].

The morphology of the fracture surface of the composites (after Charpy impact testing) was studied by SEM (see Figure 8). It is possible to identify two different morphologies: a brittle fracture and ductile fracture morphology. With regard to brittle fractures, MNA and MNA:GLYC composites (Figure 8A,F) clearly show this brittle fracture morphology. These samples, both cured without MLO co-hardener are characterized by a homogenous fractured surface, without relevant changes in surface roughness. It can also be seen that the longitudinal surfaces of individual fibers are very clean, with no adhered matrix. This kind of surface is typical of a brittle fracture during impact test, with an associated impact energy value of 20 kJ·m^−2^. Impact test-absorbed energy is directly related to the fragility of the material. However, with a more detailed analysis on these surfaces at higher magnifications (1000×) (Figure 9), it is possible to observe an improved fiber–polymer interaction, which can be seen by a reduction of the gap between lignocellulosic flax fiber and the ELO matrix, when flax fabric has been previously treated with glycidyl silane. It is possible to confirm that the interaction between the flax fiber and the surrounding polymeric ELO-based matrix has been improved, by reducing substantially the gap between them. This better fiber–matrix continuity is responsible for load/stress transfer from the matrix to the fiber. This phenomenon leads to a positive effect on mechanical performance of composites, as the Charpy’s impact energy showed. In fact, the impact-absorber energy for MNA:GLYC composite (with a previous silanization process with a glycidyl silane) is 19.6% than that of the MNA composite without any previous silane treatment).

On the other hand, Figure 8B–E show the fracture surface of composites cured with the presence of MLO as co-hardener. It is possible to observe differences with the abovementioned composites cured without MLO co-hardener. Gaps between fiber and polymeric matrix are reduced, which is a positive effect on the overall fiber–matrix continuity that allows improved load transfer. This good fiber–matrix interaction can be assessed by the presence of polymeric matrix chemically bonded to the fiber, which is not pullet out during the fracture process. As is shown in Figure 9b, the use of a previous silanization treatment on fibers improves the overall matrix–fiber interaction but does not justify the existence of these polymer residues in the fibers. In fact, in this image the fiber treated with glycidyl silane appears clean. This polymer residue located on the fiber gives a positive effect on the fiber–matrix interaction. Therefore, this could be attributed to the effects of MLO co-hardener, which must be able to interact between both the flax fiber and the ELO matrix, leading to a synergistic effect: overall flexibilization as well as increasing the interaction. As we have demonstrated previously, these samples cured with MLO co-hardener are characterized by high ductile properties, with Charpy’s impact energy above 30 kJ·m^−2^, an elongation at break over 3% than and glass transition temperature lower than 50 °C. In view of the obtained results with these flax-based composites, it seems that a pre-treatment on fibers (silanization or esterification by maleic anhydride) together the use of MLO as co-hardener play an essential role on final mechanical properties.

In order to explain the possible chemical interactions that occur with the presence of these components, Figure 10 shows a schematic representation of the chemical structures that take part in the cross-linking process and composite manufacturing. The mechanical results, as was proved previously, suggest that, like many other authors described, the use of silanes allows a better interaction between the flax fibers and the ELO-based polymeric matrix [39,61,62]. This is due to the chemical interaction between the –OH groups present on cellulose and the reactive groups present in the coupling agents.

On one hand, the addition of MLO as co-hardener in a mixture with MNA, substantially improved the ductility of composites as it was expected. As Chern et al. and Balart et al. described [59,60], this is due to the presence of long fatty acid chains in MLO structure which increase considerably ductile properties as elongation at break, maximum deflection in flexural test, or a decrease on glass transition temperature.

Unexpected results are obtained when both the MLO co-hardener and a previous surface treatment on flax fabric are combined. Composites manufactured with flax fibers previously treated and the use of MLO as a co-hardener have greater ductility than composites without previous surface treatment on flax fibers. In order to explain the possible chemical interaction, Figure 11A shows the possible reaction sites between cellulose and maleic anhydride-based hardeners. The multiple –OH groups naturally present in cellulose react with MLO and MNA by opening the maleic anhydride group, giving the corresponding esters and leading to multiple anchorage points. These reactions reduce molecular mobility and provide stiffness for composites. Figure 11B shows the possible interaction with a previous surface treatment on fibers. In this case, the hydrolyzed groups present in the silanes, directly interact with the –OH groups present in cellulose, thus blocking hydroxyl groups in cellulose. Consequently, available reactions sites for MLO and the MNA are remarkable reduced. This reduction of the potential anchor points limits the chemical interaction and, therefore, the composite materials have greater ductility. On the other hand, as Raza et al. have demonstrated [63] the use of maleinized vegetable oils provide hydrophobicity to cellulose fibers by attaching its nonpolar hydrocarbon chain on the surface of the fibers through its anhydride groups which can react to form a covalent ester bond at the interface.

## 4. Conclusions

In this work, a new high environmentally friendly composites based on linen byproducts was obtained using industrial manufacturing by resin transfer molding (RTM). The developed material has a bio-based content of about 78%. As base epoxy resin, epoxidized linseed oil (ELO) was used and composites with three plies of flax fabrics were manufactured by RTM. As hardener, a binary mixture (50:50) of methyl nadic anhydride (MNA) and maleinized linseed oil (MLO) was used to increase the overall bio-based content without compromising other properties. In general, in absence of MLO as co-hardener, composites with high mechanical resistant properties are obtained due to the rigid MNA molecule. On the other hand, when MLO is used as co-hardener, the presence of long fatty acid chains provided improved ductile properties in composites materials (a percentage increase of 42% and 43% in elongation at break and Charpy’s impact energy respectively). Nevertheless, an unexpected behavior was observed due to combined effects of using MLO as co-hardener and using previous surface treatments of flax fabrics. In this case, composites show more flexibility than composites without previous surface treatment (with a percentage increase in Charpy’s impact energy and elongation at break of 26% and 87% in respectively). A previous surface treatment on flax fabrics using silanes or maleic anhydride leads to reduce the reactions sites between the ELO-based epoxy resin and the flax fiber. SEM images demonstrated how presence of MLO and treated fibers provide better fiber–matrix interaction. On the other hand, if a previous surface treatment is not carried out, the –OH groups present in the cellulose provide enough active points to create chemical bonds with MLO used as co-hardener resulting in stiffer materials, as tensile, flexural, Charpy impact test and DMA studies demonstrated. Results also suggest that the use of maleic anhydride gives the composite materials greater ductility due to esterification of cellulose surface. By selecting the amount of MLO it is possible to tailor the desired properties in a high environmental efficiency composite material, thus giving new opportunities to these green materials in a wide range of industrial applications.

## Figures and Tables

**Figure 1 polymers-11-00301-f001:**
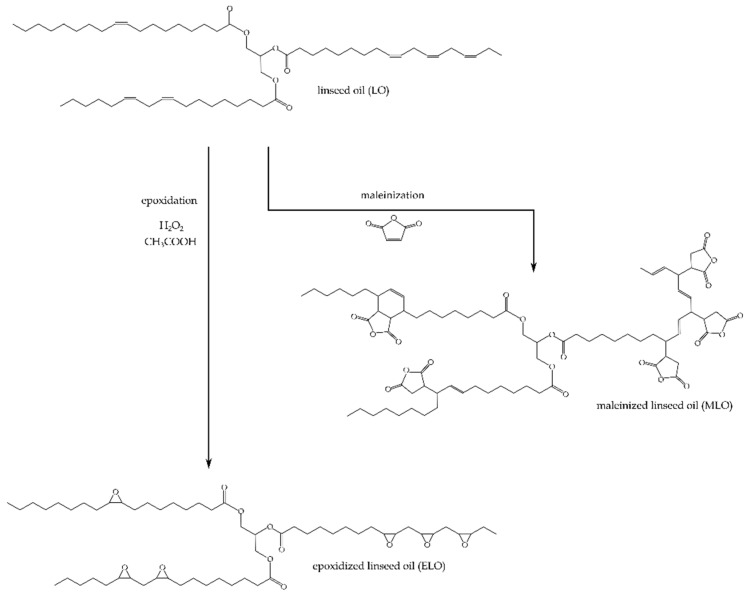
Chemical pathways used to obtain the base epoxy (epoxidized linseed oil, ELO) and the bio-based cross-linker (maleinized linseed oil, MLO).

**Figure 2 polymers-11-00301-f002:**
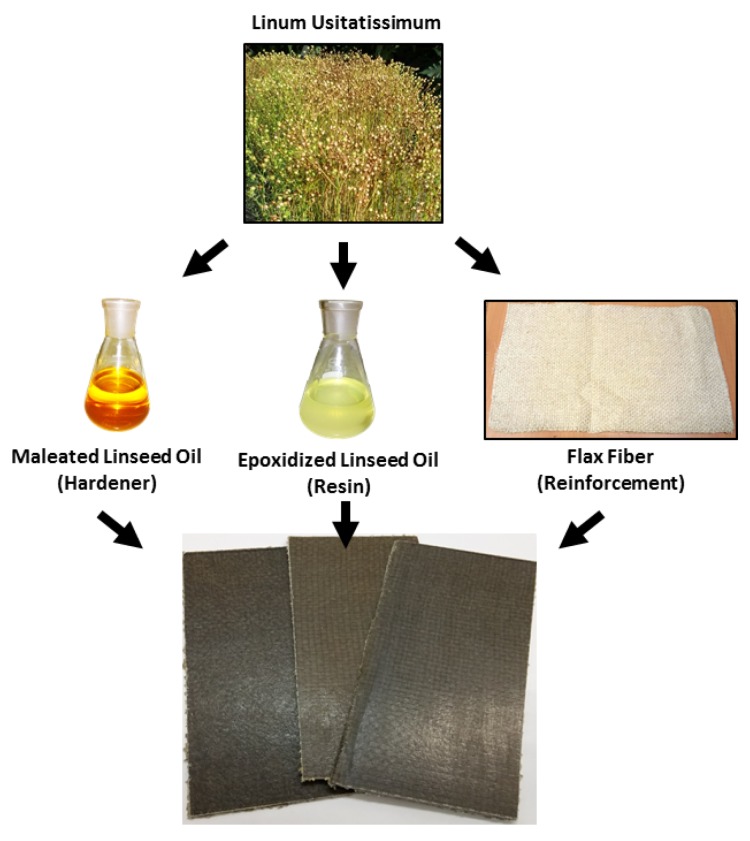
Multiple applications of *linun usitatissimum* in composite manufacturing.

**Figure 3 polymers-11-00301-f003:**
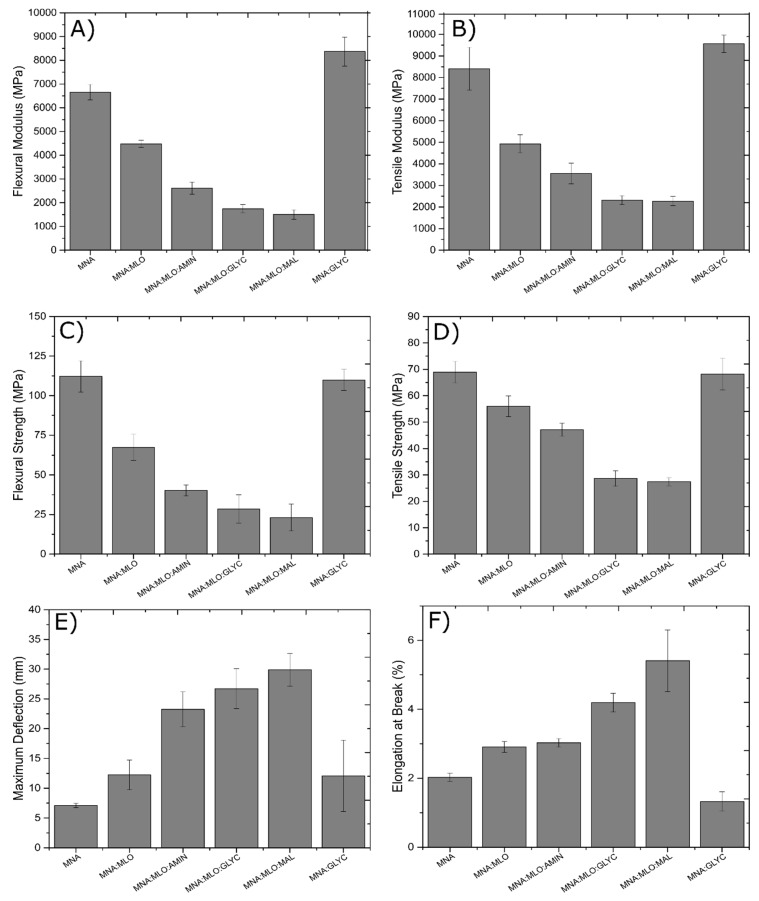
Comparative bar plot of the flexural and tensile properties of composites based on ELO epoxy resin cross-linked with MNA/MLO mixture and flax fabric reinforcement with different surface treatments: (**A**) flexural modulus, (**B**) tensile modulus, (**C**) flexural strength, (**D**) tensile strength, (**E**) maximum flexural deflection, and (**F**) % elongation at break.

**Figure 4 polymers-11-00301-f004:**
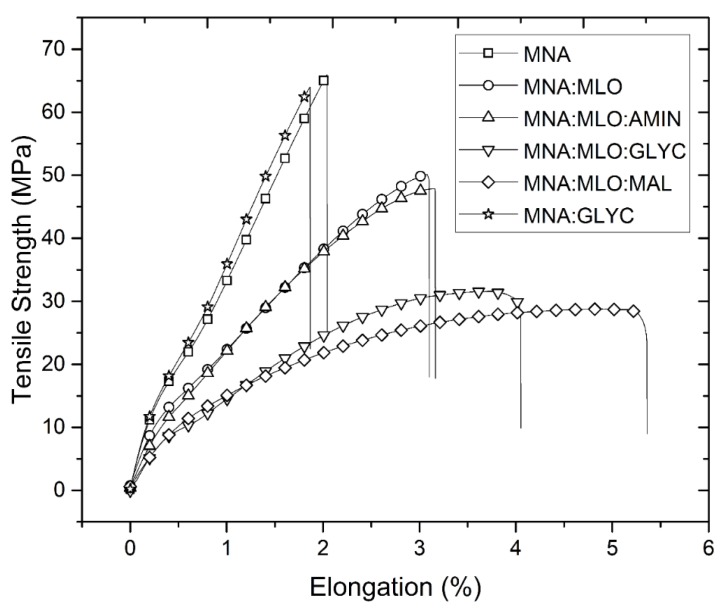
Plot representation of tensile strength (MPa) vs. Elongation.

**Figure 5 polymers-11-00301-f005:**
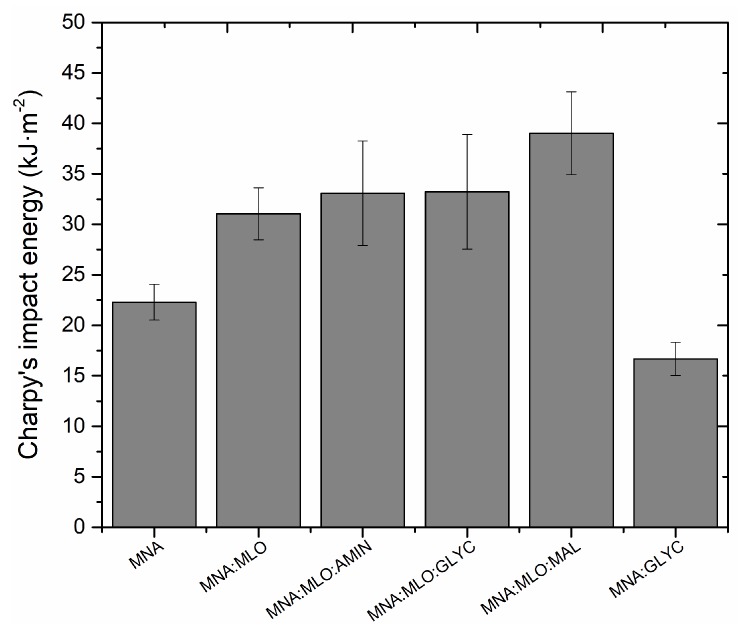
Variation of the impact energy (Charpy test) of composites based on ELO epoxy resin cross-linked with MNA/MLO mixture and flax fabric reinforcement with different surface treatments.

**Figure 6 polymers-11-00301-f006:**
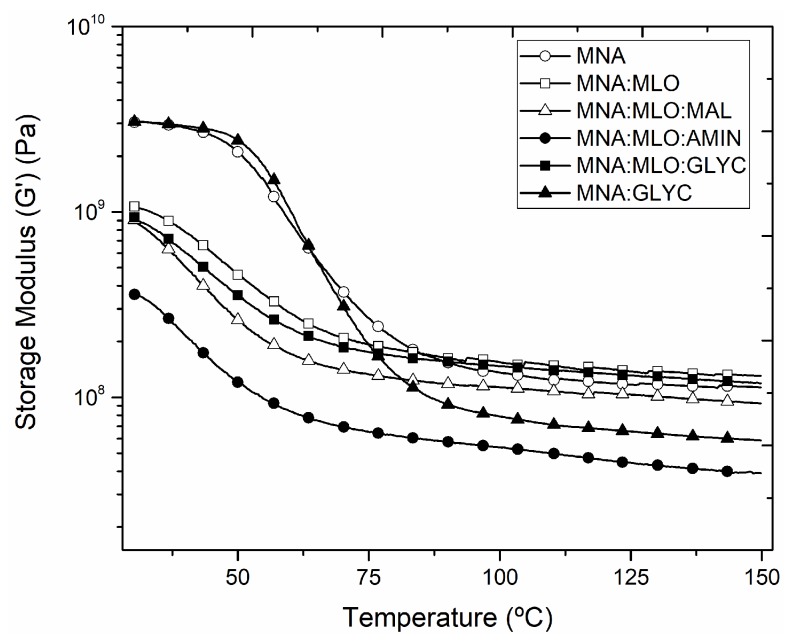
Comparative plot of storage modulus (G’) of different composites based on ELO epoxy resin cross-linked with MNA/MLO mixture and flax fabric reinforcement with different surface treatments.

**Figure 7 polymers-11-00301-f007:**
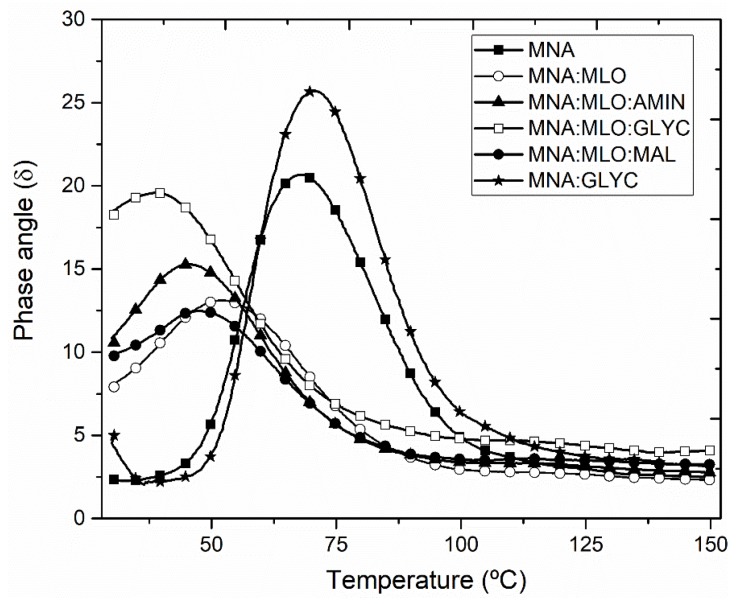
Comparative plot of the phase angle (δ) of different composites based on ELO epoxy resin cross-linked with MNA/MLO mixture and flax fabric reinforcement with different surface treatments.

**Figure 8 polymers-11-00301-f008:**
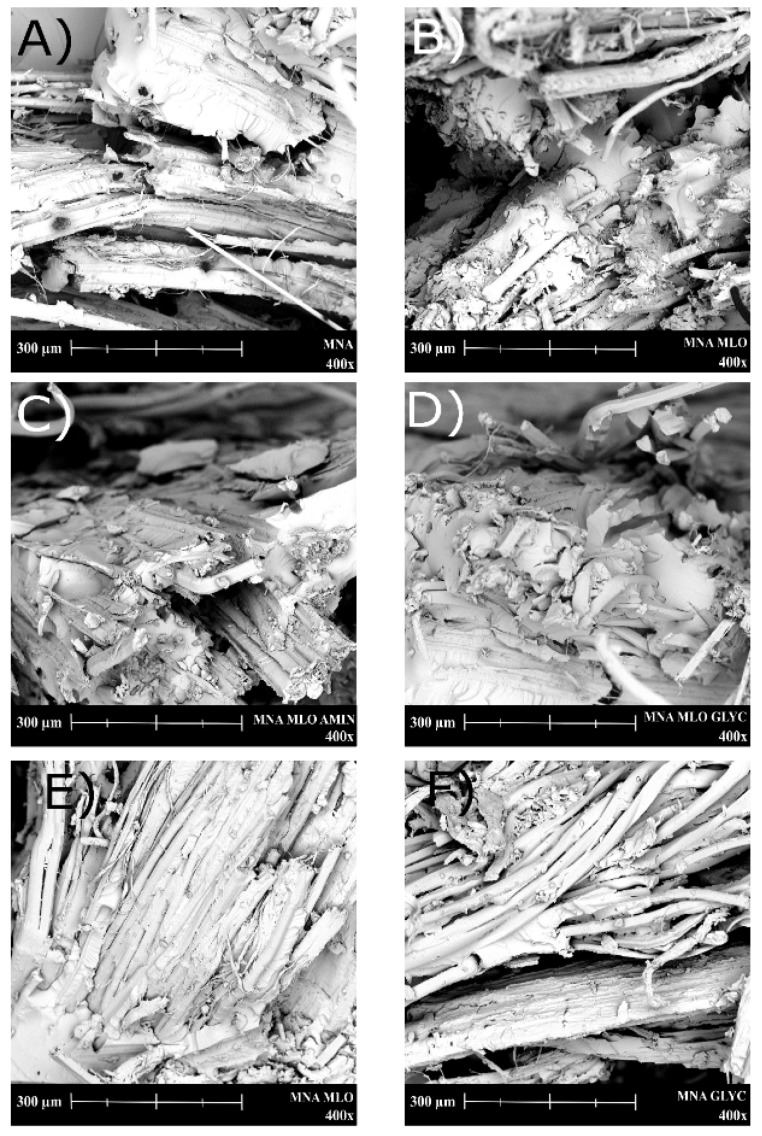
SEM micrographs (400×) of the fracture surfaces of composites based on ELO epoxy resin cross-linked with MNA/MLO mixture and flax fabric reinforcement with different surface treatments, (**A**) MNA, (**B**) MNA:MLO, (**C**) MNA:MLO:AMIN, (**D**) MNA:MLO:GLYC, (**E**) MNA:MLO:MAL, (**F**) MNA:GLYC.

**Figure 9 polymers-11-00301-f009:**
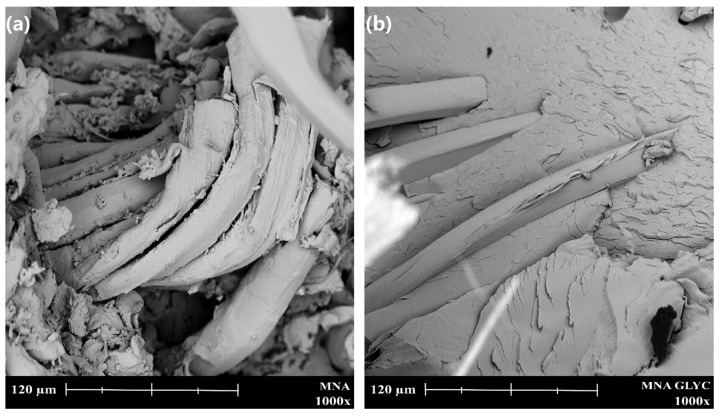
SEM micrographs (1000×) comparing the gap along flax fiber in ELO-based composites cross-linked with (**a**) MNA and no previous surface treatment and (**b**) cross-linked with MNA + previous silanization with glycidyl silane on flax fabric.

**Figure 10 polymers-11-00301-f010:**
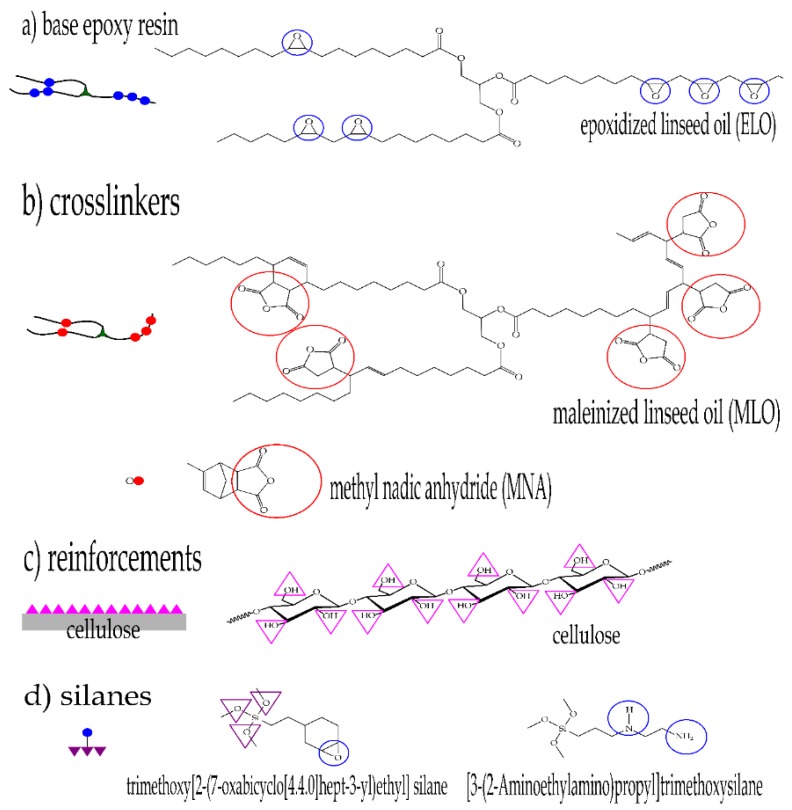
Chemical structure and schematic representation of all components in ELO-based composites with MNA/MLO hardener and different surface treatments on cellulose from flax fiber.

**Figure 11 polymers-11-00301-f011:**
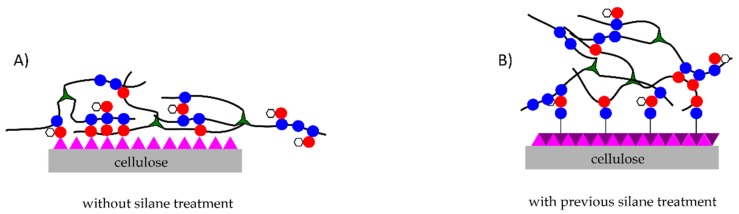
Potential chemical reactions that take place during manufacturing ELO-based composites (**A**) without previous silane treatment and (**B**) with a previous silanization surface treatment.

**Table 1 polymers-11-00301-t001:** Comparison between natural flax fiber and glass fiber. Adapted from [23].

Parameter	E-Glass	Flax Fiber
Density (g cm^−3^)	2.5	1.4
E-modulus (GPa)	76	50–70
Specific E-modulus (GPa cm^3^ g^−1^)	30	36–50
Tensile strength (GPa)	1.4–2.5	0.5–1.5
Specific tensile strength (GPa cm^3^ g^−1^)	0.5–1	0.4–1.1
Renewability	No	Yes
Recyclability	No	Yes
Disposal	Biodegradable	Not Biodegradable
Health risk when inhaled	Yes	No

**Table 2 polymers-11-00301-t002:** Composition and coding of the flax fabric composites with epoxidized linseed oil (ELO) matrix and a mixture of maleinized linseed oil (MLO) and methyl nadic anhydride (MNA) as hardeners.

No. Sample	Epoxy Resin	Hardener	Surface Treatment	Flax Fabrics Layers	Code
1	ELO	MNA	-	3	MNA
2	ELO	MNA:MLO	-	3	MNA:MLO
3	ELO	MNA:MLO	Amine silane	3	MNA:MLO:AMIN
4	ELO	MNA:MLO	Glycidyl silane	3	MNA:MLO:GLYC
5	ELO	MNA:MLO	Maleic anydride	3	MNA:MLO:MAL
6	ELO	MNA	Glycidyl Silane	3	MNA:GLYC

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
