# Peer review of "Maleinized Linseed Oil as Epoxy Resin Hardener for Composites with High Bio Content Obtained from Linen Byproducts"

_polymers, 2019, doi:10.3390/polym11020301_

Round 1

Reviewer 1 Report

In this work authors have developed novel thermosetting composites,, by means of resin transfer molding (RTM), with high green character (around 80%). Authors used a biobased epoxy matrix (epoxidized linseed oil, ELO) reinforced with flax fibers and as a hardener a liquid mixture (50 wt%:50 wt%) from methyl nadic anhydride (MNA) and maleinized linseed oil (MLO). The topic is very well related with the scope of the journal and it is interesting for both, academics and for the industrial sector, and of sure for the wide readers of Polymers. The manuscript shows logical train of thought. It is very complete work, since it involves chemical modifications, composites development, materials surfaces treatments as well as a fully materials characterization. Moreover, the obtained results are relevant and well discussed. Thus, I have no dubdt to recomend the publication of this manuscript.

I only have some minor comments and suggestions:

Introduction section:

Authors comments that Canada, Russia and China are the largest worldwide producers of flax, while Europe produce around 120,000 tons in 2007. It should be interest to update these quantities if authors have the information of the produced amount of flax in the last years.

Results

The quality of Figures 7 and 8 (SEM images) should be improved, since it is not possible to distinguish the scale bar values.

The quality of image 9 should be improved due to the fact that in the pdf version the words cannot be read.

Formal aspects:

line 340: impact energy value of 20 kJ m-2. -2 should be superscript

line 364: impact energy above 30 kJ m-2. -2 should be superscript

Author Response

Thank you very much for your e-mail dated on 05 February. We are sending the revised version of the manuscript. All suggestions made by the reviewers have been considered. Changes done to manuscript have been emphasized in yellow, in order to facilitate its searching. We have worked in accordance with all reviewer comments on the final version of the manuscript, so we consider that the version we are sending to you includes all necessary changes.

Following the suggestion of the reviewers, new figures have been added, enhancing information about surface and mechanical characterization of polymers used in this investigations. News references have been cited with the main objective to provide objective information and not based on conjectures.

Sincerely,

Vicent Fombuena

Reviewer 2 Report

This work is interesting from a technical point of view and should be published after minor revision, having addressed the issues below:

-Figures (such as 9) need to be of proper quality

-How is the functionalisation confirmed and in what extent (%)?

-Stress strain curves should be provided

-Contact angle/wetting assessment of the fibers to the used resing could strengthen the discussion on mechanism (also fracture analysis)

Author Response

(The authors gave the same response as above.)

Reviewer 3 Report

The manuscript entitled "Maleinized linseed oil as hardener for composites with high bio content obtained from linen byproducts" reports about an interesting study on the fabrication and the caracterization of bio-based composite materials potentially suitable for a wide range of industrial applications.

In my opinion, the manuscript deserves to be published on Polymers, since the experimental work is well-designed and the discussion of the obtained results is properly conducted.

However, the following concernes have to be solved:

On page 9, lines 301-305, the authors attributed to a "stabilizing effect" due to the presence of MLO, the lower decrease of G' as a function of temperature for MLO-containing composites with respect to the systems without MLO. In my opinion, the authors should better specify and describe this explaination.

Please, improve the quality of Figure 9.

Author Response

(The authors gave the same response as above.)
